# Role of the redox state of the Pirin-bound cofactor on interaction with the master regulators of inflammation and other pathways

**Tamim Ahsan**[1☯], **Sabrina Samad Shoily**[2☯], **Tasnim Ahmed**[2], **Abu Ashfaqur Sajib**[2]*

1 Molecular Biotechnology Division, National Institute of Biotechnology, Savar, Dhaka, Bangladesh,
2 Department of Genetic Engineering & Biotechnology, University of Dhaka, Dhaka, Bangladesh

☯ These authors contributed equally to this work.
* abu.sajib@du.ac.bd

**Data Availability Statement:** All relevant data are within the paper and its Supporting information files.

## Abstract

Persistent cellular stress induced perpetuation and uncontrolled amplification of inflammatory response results in a shift from tissue repair toward collateral damage, significant alterations of tissue functions, and derangements of homeostasis which in turn can lead to a large number of acute and chronic pathological conditions, such as chronic heart failure, atherosclerosis, myocardial infarction, neurodegenerative diseases, diabetes, rheumatoid arthritis, and cancer. Keeping the vital role of balanced inflammation in maintaining tissue integrity in mind, the way to combating inflammatory diseases may be through identification and characterization of mediators of inflammation that can be targeted without hampering normal body function. Pirin (PIR) is a non-heme iron containing protein having two different conformations depending on the oxidation state of the iron. Through exploration of the Pirin interactome and using molecular docking approaches, we identified that the $Fe^{2+}$-bound Pirin directly interacts with BCL3, NFKBIA, NFIX and SMAD9 with more resemblance to the native binding pose and higher affinity than the $Fe^{3+}$-bound form. In addition, Pirin appears to have a function in the regulation of inflammation, the transition between the canonical and non-canonical NF-κB pathways, and the remodeling of the actin cytoskeleton. Moreover, Pirin signaling appears to have a critical role in tumor invasion and metastasis, as well as metabolic and neuro-pathological complications. There are regulatory variants in *PIR* that can influence expression of not only *PIR* but also other genes, including *VEGFD* and *ACE2*. Disparity exists between South Asian and European populations in the frequencies of variant alleles at some of these regulatory loci that may lead to differential occurrence of Pirin-mediated pathogenic conditions.

## Introduction

The immune system responds to stress or harmful stimuli, such as toxic compounds or pathogens, by initiating a cascade of inflammation, and thus restores homeostasis [1, 2]. However,

**Funding:** This study was supported by a grant under the Special Allocation in Science and Technology from the Ministry of Science and Technology, Bangladesh. The funders had no role in study design, data collection and analysis, decision to publish, or preparation of the manuscript.

**Competing interests:** The authors have declared that no competing interests exist.

pathological fueling of certain pathways and their systemic effects through interconnected networks of biological pathways and processes plays a key role in the onset or progression of immune-mediated inflammatory diseases and leads to the emergence of multimorbidity [2, 3]. Abnormal regulation of one or more of the three key inflammatory pathways- Nuclear factor kappa B (NF-κB), Mitogen-activated protein kinase (MAPK), and Janus kinase/signal transducer and activator of transcription (JAK-STAT), results in progression to inflammation-mediated diseases [1].

Apparently, inhibition of NF-κβ may provide effective treatment option for diseases associated with its overactivation. However, as NF-κB transcription factors regulate the expression of genes involved in several critical physiological processes, including cell survival, growth, proliferation, oxidative stress responses, inhibition of apoptosis, inflammation and immune responses, direct inhibition of NF-κB signaling can lead to immunodeficiency and dysregulation of the associated physiological activities [4, 5]. Thus, bypass routes of suppressing over-activation of the NF-κB pathway can be the choice of treatment for chronic inflammation-mediated diseases.

Pirin (encoded by the gene *PIR*) is a highly conserved 290 amino acid long and 32 kDa nuclear protein. Pirin was discovered as an interactor of nuclear factor I/CCAAT box transcription factor (NFI/CTF1) in a yeast two-hybrid (Y2H) screening [6]. On the basis of sequence and structural homology, this protein, which consists of two antiparallel β-barrel domains with an iron cofactor located within the N-terminal domain, belongs to the functionally diverse cupin superfamily [7, 8]. Depending on the oxidation state of the bound iron ($Fe^{2+}$ and $Fe^{3+}$), there are two different conformations of Pirin which vary in their R-shaped surface area [9]. Pirin is detected at low levels in all human tissues, but the transcript levels are the highest in the heart and liver [6, 10]. *PIR* orthologs have been found in prokaryotic organisms, fungi, plants and other mammals [8]. Both human and bacterial Pirins were demonstrated to be functionally similar to quercetin 2,3-dioxygenase, which can use quercetin flavonoid as a substrate. This function of Pirin can be inhibited by the addition of the inhibitors of quercetin 2,3-dioxygenase [11].

Pirin is one of the regulators of NF-κB signaling in the nucleus, and this regulation is controlled by its iron center [9, 12]. Under oxidizing conditions, the Fe(III) form of Pirin binds transcription factor NF-κB p65 and enhances the interaction between DNA and NF-κB p65 [9, 13]. Additionally, Pirin coregulates the NF-κB transcription pathway through interaction with the oncoprotein B-cell CLL/lymphoma 3 (BCL3) [8]. In spite of being a member of the Inhibitor of Kappa-B (IκB) family of proteins, which contribute to repression of the NF-κB signaling cascade, BCL3 exerts both transactivation and transrepressor activities in the regulation of NF-κB associated pathways [14]. Thus, characterization of the molecular association of Pirin with NF-κB and the consequences of this interaction may reveal new targets for inhibition of pathogenic inflammation.

Over-expression of *PIR* in breast cancer cells play roles in tumorigenesis and cancer progression through induction of the E2F1 pathway [15]. Pirin was also demonstrated to be associated with metastasis of melanoma and cervical cancer cells [16, 17]. Abnormal pattern of sub-cellular localization of Pirin was observed in a subset of melanomas [7]. The potential role of Pirin regulated pathways in cancer progression and/or metastasis may be elucidated by analysis of its interaction with other proteins and associated biological processes. Variants in the *PIR* gene have been linked to interethnic disparities in COVID-19 case fatality rates [18].

Although established as a regulator of inflammation [9], there are barely any studies solely focusing on the pathways of Pirin and its role in disease pathogenesis. The existing databases lack information regarding the regulatory and functional roles of Pirin. Understanding how a protein interacts with other proteins in a protein-protein interaction (PPI) network is

important for figuring out its overall role and how it controls other proteins [19, 20]. In this study, we have utilized protein-protein interactions and network-based methods to functionally characterize Pirin. We also explored the potentially pathogenic variants that may modulate *PIR* expression and their frequencies in populations worldwide.

## Materials and methods

### Determination of interaction partners of Pirin

Protein-protein interaction (PPI) databases- BioGRID (v4.4.201) [21], IMEx [22], IntAct (v1.0.2) [23], MINT [24], STRING (v11.5) [25] and Mentha [26] were searched with the term "PIR" to retrieve interacting partners of Pirin as well as evidence of the interactions. The Biological General Repository for Interaction Datasets (BioGRID) is an open access biomedical repository that houses comprehensively curated protein, genetic, and chemical interactions as well as post-translational modifications for humans and all major model organism species [21]. The IntAct molecular interaction database populates data derived from literature curation or direct user deposition, and it actively contributes to the International Molecular Exchange Consortium (IMEx) partners via a sophisticated web-based curation platform [23, 27]. Mentha is a PPI resource that brings together protein interactions from primary databases that are in compliance with IMEx curation policies [26]. The Molecular INTeraction database (MINT) is a public PPI database that stores molecular interactions that have been reported in peer-reviewed journals [24]. The Search Tool for Retrieval of Interacting Genes/Proteins (STRING) database integrates known as well as predicted interactions, including both physical (direct) and functional (indirect) associations [25, 28].

### Analysis of binding between Pirin and its direct interactors

In this study, Y2H system was considered as the experimental evidence of direct PPI, as this genetic system enables the detection of direct interaction between proteins [29, 30]. Based on this criterion BCL3, nuclear factor of kappa light polypeptide gene enhancer in B-cells inhibitor alpha (NFKBIA, also known as IKBA), nuclear factor I/X (NFIX), and SMAD family member 9 (SMAD9) were identified as the direct interactors of Pirin (Table 1). The domain architectures of the direct interactors were obtained from either the UniProt Knowledgebase (UniProt-KB) or Pfam 35.0 database (depending on availability) [31, 32]. X-ray crystallographic structures of BCL3 (PDB ID: 1K1A), NFKBIA (PDB ID: 1IKN), and SMAD9 (PDB ID: 6FZT) were retrieved from the Protein Data Bank (PDB) [33]. Since crystallographic structure of NFIX was not available, its predicted structure was downloaded from AlphaFold2 [34]. Although X-ray crystallographic structures of both the $Fe^{2+}$ (PDB code: 1J1L) and $Fe^{3+}$ bound (PDB code: 4GUL) conformations of Pirin are available [35], the crystal structure of the $Fe^{2+}$ bound Pirin contained six modified methionine residues. To maintain uniformity, structures of both the conformations were modeled using template-based modeling at SWISS-MODEL server [36]. Amino acid sequence of Pirin (UniProt accession: O00625-1) was collected from UniProt [31].

To get insight into the association of direct interaction partners with Pirin, the putative interacting domains of the direct interactors were docked to modeled structures of both ferrous ($Fe^{2+}$) and ferric ($Fe^{3+}$) conformations of Pirin protein using the HawkDock web server [37]. To calculate the binding free energies, Molecular Mechanics/Generalized Born Surface Area (MM/GBSA) analysis was performed on the HawkDock server [37, 38].

The top 10 docked models were re-ranked using the DOCKSCORE web server [39]. In DOCKSCORE, weights are assigned to multiple parameters of the putative interface between the two protein partners, including surface area, short contacts, evolutionary conservation, spatial clustering as well as the presence of positively charged and hydrophobic residues. By

**Table 1. Interaction partners of Pirin protein.**

| Interactor | Description | Evidence | Sources of evidence |
|---|---|---|---|
| ABL2 | Tyrosine-protein kinase ABL2 | Database annotation, Automated text-mining | STRING |
| ACAT2 | Acetyl-CoA acetyltransferase 2 | Co-fractionation | BioGRID, Mentha |
| ADK | Adenosine kinase | Co-fractionation | BioGRID, Mentha |
| AGR2 | Anterior gradient 2 | Proximity label-MS | BioGRID |
| ANXA7 | Annexin A7 | Co-fractionation | BioGRID, Mentha |
| ARHGDIA | Rho GDP dissociation inhibitor alpha | Co-fractionation | BioGRID, Mentha |
| ASS1 | Argininosuccinate synthase 1 | Co-fractionation | BioGRID, Mentha |
| BCAR1 | Breast cancer anti-estrogen resistance 1 | Affinity capture-MS | BioGRID |
| **BCL3** | B-cell lymphoma 3 | Affinity capture-Western, Reconstituted complex, Y2H | BioGRID, IMEx, IntAct, Mentha, MINT |
| C11orf54 | Chromosome 11 open reading frame 54 | Co-fractionation | BioGRID, Mentha |
| C2orf68 | Chromosome 2 open reading frame 68 | Affinity capture-MS | BioGRID, IMEx, IntAct, Mentha |
| CAPG | Capping actin protein, gelsolin-like | Co-fractionation | BioGRID, Mentha |
| CFL1 | Cofilin 1 (non-muscle) | Co-fractionation | BioGRID, Mentha |
| CFL2 | Cofilin 2 (muscle) | Co-fractionation | BioGRID, Mentha |
| CLNS1A | Chloride nucleotide-sensitive channel 1A | Co-fractionation | BioGRID, Mentha |
| CTF1 | Cardiotrophin-1 | Automated text-mining | STRING |
| DAZAP1 | DAZ associated protein 1 | Co-fractionation | BioGRID, Mentha |
| DCXR | Dicarbonyl/L-xylulose reductase | Co-fractionation | BioGRID, Mentha |
| DDX39A | DEAD (Asp-Glu-Ala-Asp) box polypeptide 39A | Affinity capture-RNA | BioGRID |
| DSTN | Destrin (actin depolymerizing factor) | Co-fractionation | BioGRID, Mentha |
| FAHD1 | Fumarylacetoacetate hydrolase domain containing 1 | Co-fractionation | BioGRID, Mentha |
| FGB | Fibrinogen beta chain | Affinity capture-MS | BioGRID, IMEx, IntAct, Mentha |
| FH | Fumarate hydratase | Co-fractionation | BioGRID, Mentha |
| FUBP1 | Far upstream element (FUSE) binding protein 1 | Co-fractionation | BioGRID, Mentha |
| GNB2L1 (RACK1) | Guanine nucleotide binding protein (G protein), beta polypeptide 2-like 1 (Receptor For Activated C Kinase) | Co-fractionation | BioGRID, Mentha |
| GNPDA1 | Glucosamine-6-phosphate deaminase 1 | Co-fractionation | BioGRID, Mentha |
| GNPDA2 | Glucosamine-6-phosphate deaminase 2 | Co-fractionation | BioGRID, Mentha |
| GOT1 | Glutamic-oxaloacetic transaminase 1, soluble | Co-fractionation | BioGRID, Mentha |
| GRHPR | Glyoxylate reductase/ hydroxypyruvate reductase | Co-fractionation | BioGRID, Mentha |
| GSR | Glutathione reductase | Co-fractionation | BioGRID, Mentha |
| HMGCL | 3-hydroxymethyl-3-methylglutaryl-CoA lyase | Co-fractionation | BioGRID, Mentha |
| HSPE1 | Heat shock 10kDa protein 1 | Co-fractionation | BioGRID, Mentha |
| HTRA2 | HtrA serine peptidase 2 | Co-fractionation | BioGRID, Mentha |
| KHDRBS1 | KH domain containing, RNA binding, signal transduction associated 1 | Proximity label-MS | BioGRID, IMEx, IntAct |
| KLHL20 | Kelch-like family member 20 | Affinity capture-MS | BioGRID, IMEx, IntAct, Mentha |
| MIF | Macrophage migration inhibitory factor (glycosylation-inhibiting factor) | Co-fractionation | BioGRID, Mentha |
| NANS | N-acetylneuraminic acid synthase | Co-fractionation | BioGRID, Mentha |
| NCK1 | Cytoplasmic protein NCK1 | Database annotation | STRING |
| NCKAP1 | Nck-associated protein 1 | Database annotation | STRING |
| **NFIX** | Nuclear factor I/X (CCAAT-binding transcription factor) | Y2H | BioGRID, Mentha |
| **NFKBIA** | Nuclear factor of kappa light polypeptide gene enhancer in B-cells inhibitor, alpha | Y2H | BioGRID, Mentha |
| NXF1 | Nuclear RNA export factor 1 | Affinity capture-RNA | BioGRID, Mentha |

(*Continued*)

**Table 1.** (Continued)

| Interactor | Description | Evidence | Sources of evidence |
|---|---|---|---|
| PARK7 | Parkinson protein 7 | Co-fractionation | BioGRID, Mentha |
| PDCD6IP | Programmed cell death 6 interacting protein | Co-fractionation | BioGRID |
| PIWIL1 | Piwi-like protein 1 | Automated text-mining | STRING |
| PIWIL2 | Piwi-like protein 2 | Automated text-mining | STRING |
| PIWIL4 | Piwi-like protein 4 | Automated text-mining | STRING |
| PPIA | Peptidylprolyl isomerase A (cyclophilin A) | Co-fractionation | BioGRID |
| PPP2CA | Protein phosphatase 2, catalytic subunit, alpha isozyme | Co-fractionation | BioGRID, Mentha |
| PSMA7 | Proteasome subunit alpha type-7 | Database annotation, Automated text-mining | STRING |
| PSMB2 | Proteasome (prosome, macropain) subunit, beta type, 2 | Co-fractionation | BioGRID, Mentha |
| PUF60 | Poly-U binding splicing factor 60KDa | Affinity capture-MS | BioGRID |
| RAC1 | Ras-related C3 botulinum toxin substrate 1 | Database annotated, Automated text-mining | STRING |
| RIN3 | Ras and Rab interactor 3 | Affinity capture-MS | BioGRID |
| SIN3A | SIN3 transcription regulator family member A | Co-fractionation | BioGRID, Mentha |
| **SMAD9** | SMAD family member 9 | Y2H | BioGRID, IntAct, MINT, Mentha |
| SNX3 | Sorting nexin 3 | Co-fractionation | BioGRID, Mentha |
| SOD1 | Superoxide dismutase 1, soluble | Co-fractionation | BioGRID, Mentha |
| SPEN | Spen family transcriptional repressor | Co-fractionation | BioGRID, Mentha |
| SRI | Sorcin | Co-fractionation | BioGRID, Mentha |
| SRXN1 | Sulfiredoxin 1 | Co-fractionation | BioGRID, Mentha |
| STAM | Signal transducing adaptor molecule (SH3 domain and ITAM motif) 1 | Co-fractionation | BioGRID, Mentha |
| SUGT1 | SGT1, suppressor of G2 allele of SKP1 (*S. cerevisiae*) | Co-fractionation | BioGRID, Mentha |
| TIA1 | TIA1 cytotoxic granule-associated RNA binding protein | Co-fractionation | BioGRID, Mentha |
| TIAL1 | TIA1 cytotoxic granule-associated RNA binding protein-like 1 | Co-fractionation | BioGRID, Mentha |
| TKT | Transketolase | Co-fractionation | BioGRID, Mentha |
| UBE2L3 | Ubiquitin-conjugating enzyme E2L 3 | Co-fractionation | BioGRID |
| UBL7 | Ubiquitin-like 7 | Co-fractionation | BioGRID, Mentha |
| WASF2 | Wiskott-Aldrich syndrome protein family member 2 | Database annotation, Automated text-mining | STRING |

considering these weights the normalized weighted score is calculated, using which a Z-score for each pose is provided to facilitate identification of native or near native pose from protein-protein docked poses [39]. The best models were chosen based on their pose, binding free energy from MM/GBSA analysis and Z-scores from DOCKSCORE. For these models, 2D interaction map was constructed using the structural analysis tool iCn3D [40]. Hotspot residues that contribute the most to binding between the proteins in these docked structures were identified using SpotOn [41]. For SMAD9 and NFIX, selected poses were superimposed on DNA-bound conformation of MH1 domains to ensure that Pirin was not interacting with DNA-binding residues. PIR-NFKBIA docked structure was superimposed on the NFKBIA-NF-κB complex (PDB ID: 1IKN) to confirm that Pirin binding site did not considerably overlap with binding sites of the other proteins in NFKBIA.

## Construction and visualization of PPI network

Using PIR and its identified interactors as input, network analysis was performed with NetworkAnalyst 3.0 [42]. The network was constructed based on the PPI data in the IMEx

database (using the default parameters and minimum network). The network was visualized with the Cytoscape (v3.8.0) software [43].

## Enrichment analysis of Pirin and its interaction partners

**Pathway enrichment.** Pathway enrichment analysis facilitates gaining mechanistic insight and makes interpretations easier by summarizing a large list of genes as a smaller list of pathways that are enriched in the determined gene list more than would be expected by chance [44]. Pathway enrichment was performed on all nodes of the minimum network through NetworkAnalyst 3.0 using each of the Kyoto Encyclopedia of Genes and Genomes (KEGG) [45] and Reactome [46] databases as the source for annotated pathways [42].

**Integrated functional network formation.** To attain an integrated view of the pathways and processes associated with Pirin signaling, a network with the KEGG and Reactome pathways as well as Gene Ontology (GO): Biological process (BP) was formed using ClueGO v2.5.8 and CluePedia v1.5.8 plug-in of Cytoscape [47, 48]. For enrichment, a two-sided hypergeometric test with Benjamini-Hochberg correction was performed, GO terms were merged, and a kappa score of 0.5 was used as the threshold value. ClueGo facilitates formation of functionally organized network of pathway terms and biological processes enriched from precompiled annotation files for a given set of genes [47]. CluePedia allows visualization of functionally connected genes and their shared pathways in a ClueGo constructed network [48].

**Disease enrichment.** Diseases associated with Pirin as well as its interactors were enriched using Enrichr [49, 50] with DisGeNET as the source database. Enrichr is a comprehensive enrichment and functional annotation web server. DisGeNET (v7.0) is a knowledge management platform that integrates and standardizes data regarding association of diseases with genes and variants from various sources [51]. The gene-disease associations (GDA) network was based on expert-curated data. Additionally, the DisGeNET (v7.0) [51] app at the Cytoscape [43] was used for visualizing associations of diseases with the direct interactors of Pirin as well as the RELA gene (encodes NF-κB p65) as binding of p65 TF to DNA is modulated by Pirin [9].

**Retrieval of regulatory variants.** *PIR* variants with potential regulatory roles were identified from rVarBase (no filters applied) [52]. Ensembl Variant Effect Predictor (VEP) [53] was used to collect the features (chromosomal location, consequences with respect to PIR and impact) of these variants. Among these, expression quantitative trait loci (eQTLs) or the variants that can influence the expression of genes in at least one tissue were identified through Genotype-Tissue Expression (GTEx) portal v8 [54]. Frequencies of the eQTLs in five superpopulations (African, admixed American, East Asian, European and South Asian) were determined from the Phase 3 haplotype data from the 1000 Genomes Project [55] using the LDhap tool at the open source LDlink suite [56].

## Result

## Interactors of Pirin protein

Through exploration of six PPI databases, 69 unique direct and indirect interaction partners of Pirin were identified (Table 1). The direct interactors of Pirin (shown in bold font in Table 1) identified by Y2H system are BCL3, NFKBIA, NFIX, and SMAD9.

## Docked structures of Pirin and its direct partners

The models chosen based on pose, binding free energy, and Z-scores of the docked structures of Pirin and its direct interaction partners as well as the residues that significantly contribute to

the interactions are shown in Figs 1–4. The binding free energy and Z-scores, respectively, from MM/GBSA and DOCKSCORE analysis of the selected models, are given in Table 2. All the domains of these interactors are given in S1 Table. All seven ankyrin repeat domains of BCL3 are required for binary interaction between Pirin and BCL3 [8]. Therefore, ankyrin repeats of BCL3 and NFKBIA were used for docking with Pirin. Such domains are not present in NFIX and SMAD9, but the Mad homology 1 (MH1) domain is commonly present in both these proteins (S1 Table). As stable interactions between proteins are mediated by domains, this common domain was selected for docking of NFIX and SMAD9 to Pirin. $Fe^{2+}$ bound conformation of Pirin appeared to be the conformation that may bind to the direct interacting partners.

## Integrated view of Pirin and its interaction partners through network building

As the first-order interaction network created by NetworkAnalyst 3.0 was very dense (with more than 2000 nodes) due to a large number of query genes (seeds), it was trimmed to a minimum network that retained only the seeds and their connecting nodes for improving performance and ease of further analysis [42, 57]. This minimum interaction network, representing comprehensive Pirin signaling, contains 141 nodes and the first degree interactors of Pirin are highlighted in pink (Fig 5). Along with BCL3, NFIX, NFKBIA and SMAD9, the other first degree interaction partners of Pirin in the network are Signal transducing adaptor molecule (STAM), Proteasome 20S subunit beta 2 (PSMB2), Chloride nucleotide-sensitive channel 1A (CLNS1A), Histone deacetylase 2 (HDAC2), Protein phosphatase 2 catalytic subunit alpha (PPP2CA), SGT1 homolog MIS12 kinetochore complex assembly cochaperone (SUGT1), Spen family transcriptional repressor (SPEN), Ubiquitin-like 7 (UBL7), and Rho GDP dissociation inhibitor alpha (ARHGDIA). All of these, except HDAC2, were query genes given as input.

## Functional analysis of Pirin and its interaction partners

**Enriched pathways and GO terms.** The statistically significant biological process or pathway terms associated with Pirin and its 69 direct and indirect interaction partners (Table 3) obtained through ClueGO analysis are shown in Fig 6. Among the pathways enriched for all

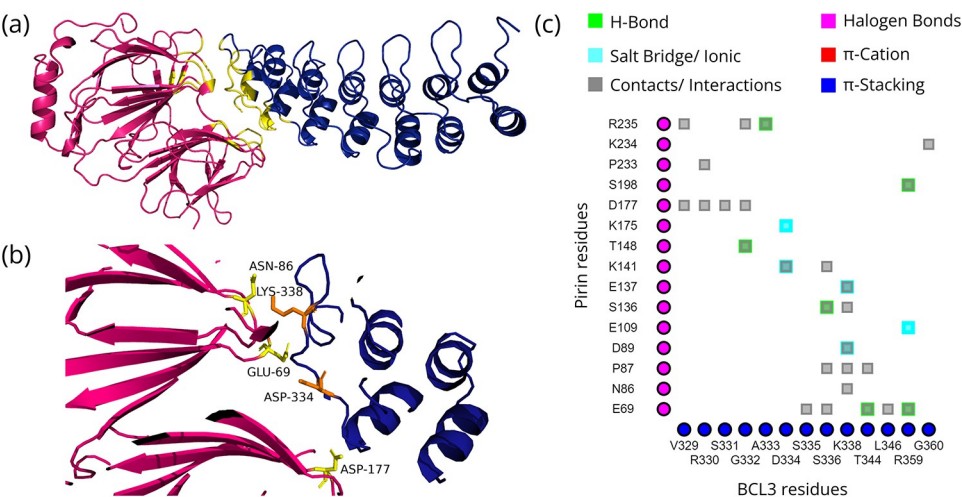

**Fig 1. Interactions between Pirin and BCL3.** (a) Ankyrin repeats of BCL3 (blue) docked to Pirin (hotpink) with interface shown in yellow; (b) Hotspot (Pirin residues = yellow, partner residues = orange) within the interface; and (c) 2D interaction map where the vertical and horizontal axis represent the Pirin and BCL3 residues, respectively.

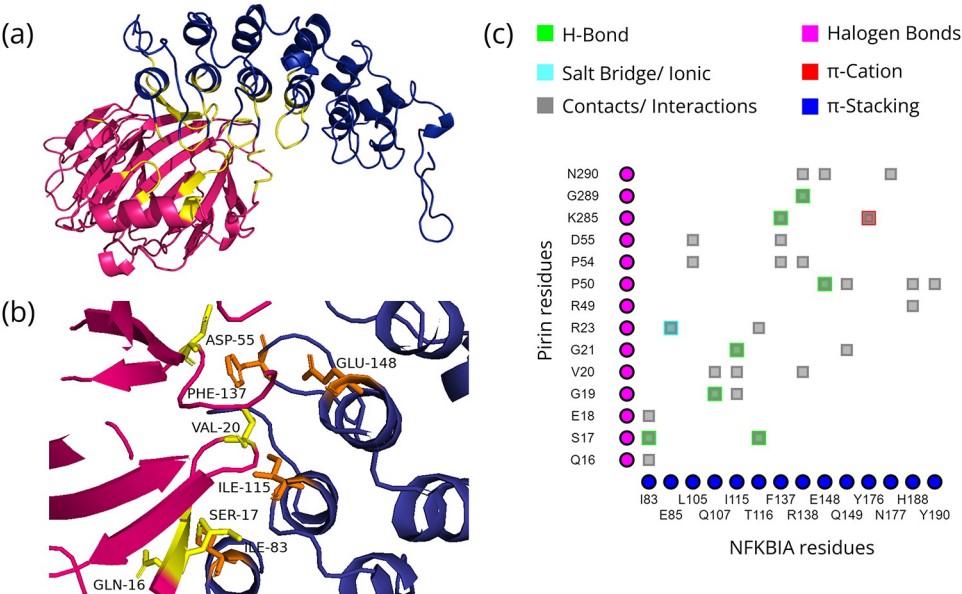

**Fig 2. Interactions between Pirin and NFKBIA.** (a) Ankyrin repeats of NFKBIA (blue) docked to Pirin (hotpink) with interface shown in yellow; (b) Hotspot (Pirin residues = yellow, partner residues = orange) within the interface; and (c) 2D interaction map where the vertical and horizontal axis represent the Pirin and NFKBIA residues, respectively.

141 genes (nodes) of the minimum network formed using Pirin and its interaction partners as input, top 20 based on the KEGG and Reactome databases are given in Fig 7.

The pathway involving activation of Wiskott-Aldrich syndrome protein (WASP) and WASP-family verprolin-homologous protein (WAVE) by the Rho family of GTPases (RHO GTPase) is most significantly associated with Pirin and its interaction partners (Fig 6). The

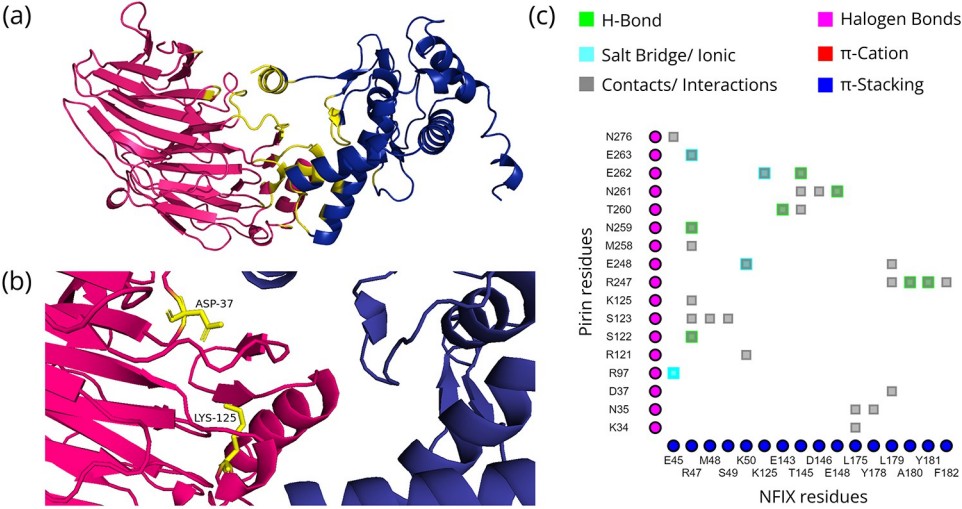

**Fig 3. Interactions between Pirin and NFIX.** (a) MH1 domain of NFIX (blue) docked to Pirin (hotpink) with interface shown in yellow; (b) Hotspot (Pirin residues = yellow, partner residues = orange) within the interface; and (c) 2D interaction map where the vertical and horizontal axis represent the Pirin and NFIX residues, respectively.

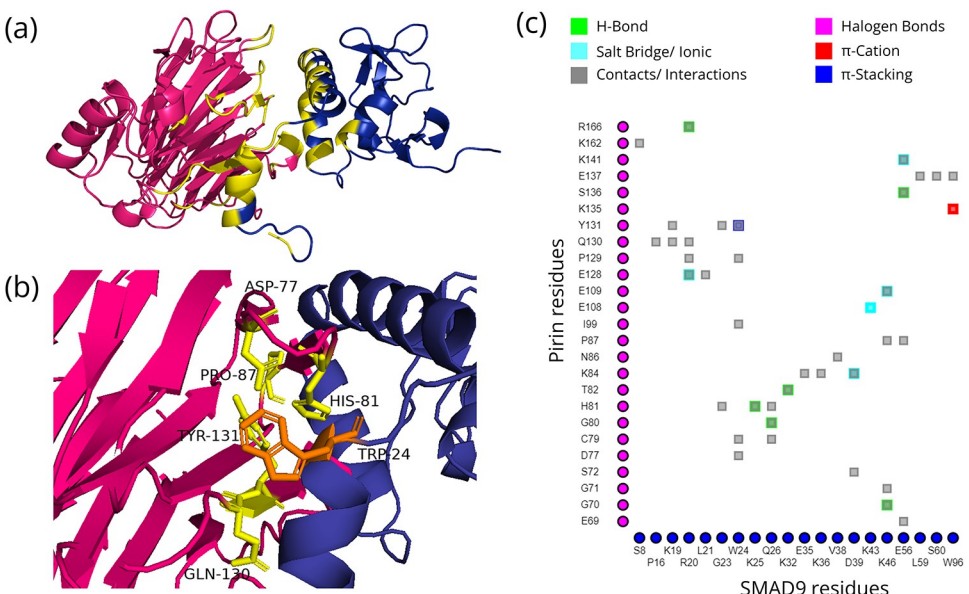

**Fig 4. Interactions between Pirin and SMAD9.** (a) MH1 domain of SMAD9 (blue) docked to Pirin with interface shown in yellow; (b) Hotspot (Pirin residues = yellow, partner residues = orange) within the interface; and (c) 2D interaction map where the vertical and horizontal axis represent the Pirin and SMAD9 residues, respectively.

**Table 2. Binding free energies and z-scores of the selected models.**

| Interaction partner | Pirin conformation | Interacting domain/ repeats of partner | HawkDawk model | HawkDock binding free energy (kcal/mol) | DOCKSCORE Z-score |
|---|---|---|---|---|---|
| BCL3 | $Fe^{2+}$ | Ankyrin repeats | Model 3 | -28.38 | 1.1034 |
| NFKBIA | $Fe^{2+}$ | Ankyrin repeats | Model 3 | -28.05 | 1.0903 |
| NFIX | $Fe^{2+}$ | MH1 domain | Model 2 | -37.69 | 0.8811 |
| SMAD9 | $Fe^{2+}$ | MH1 domain | Model 7 | -22.11 | 1.0015 |

correlation of Pirin signaling with actin filament fragmentation, regulation of actin dynamics for phagocytic cup formation, signaling by the B cell receptor (BCR), activation of NF-κB in B cells, gene and protein expression by JAK-STAT signaling after interleukin-12 (IL12) stimulation, nucleotide sugar biosynthetic process, metabolism of tyrosine and pyruvate, signaling by PTK6 and FGFR2, regulation of intrinsic apoptotic signaling pathway, regulation of DNA binding, and negative regulation of transposition is significantly highlighted by the ClueGo integrative analysis.

In KEGG pathway enrichment (Fig 7a) for the extended Pirin interactome (Pirin, its interaction partners and further nodes required to connect the seed nodes in minimum network (Fig 5)), the "ErbB signaling pathway" is the most significantly over-represented (q value < 0.005). ErbB signaling pathway influences cell proliferation, survival, differentiation, and migration (KEGG pathway entry: hsa04012). ErbB family of receptor tyrosine kinases (RTKs) comprises four distinct receptors: the epidermal growth factor receptor (EGFR/ErbB1/HER1), ErbB2 (neu, HER2), ErbB3 (HER3) and ErbB4 (HER4) [58]. Alteration of the pathways regulated by these receptors play key role in development of multiplicity of cancers as well as invasion of tumor cells [58, 59]. The highest number of genes (n = 18) are mapped to "pathways in cancer" in the KEGG database (Fig 7a). Node genes were also clustered in other

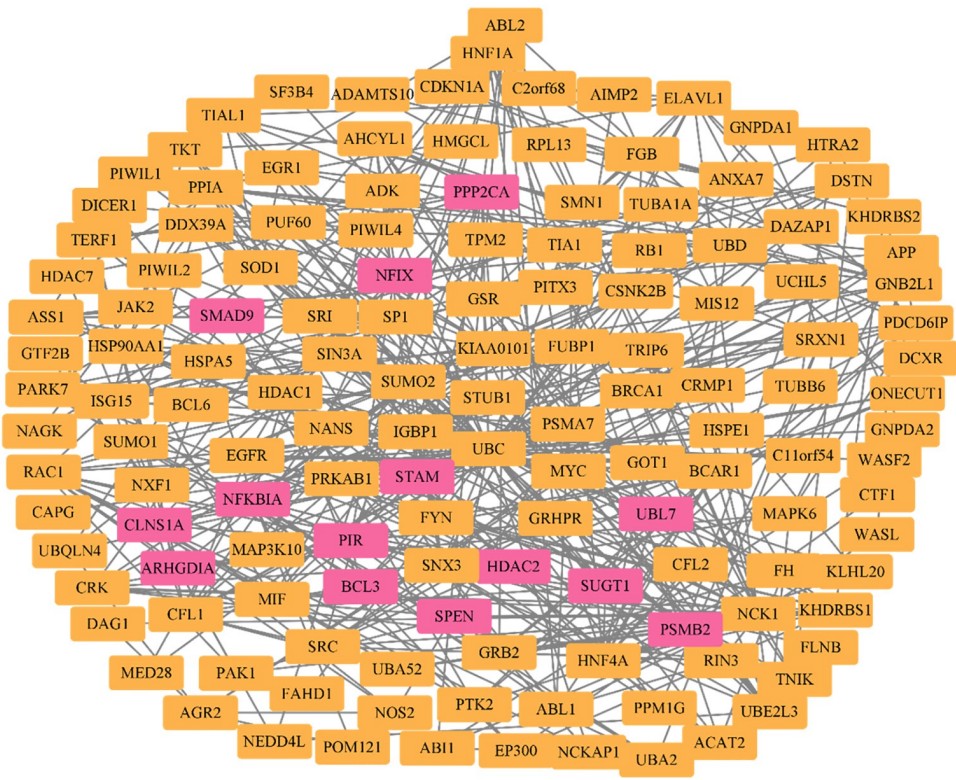

**Fig 5. PPI network among Pirin and its interaction partners (represented by gene symbols).** First degree
interaction partners of Pirin (undirected) are highlighted in pink.

carcinogenic pathways, including chronic myeloid leukemia (CML), renal cell carcinoma,
viral carcinogenesis, bladder cancer, and prostate cancer. Endocrine resistance is also enriched
which is a major clinical issue encountered in treating metastatic hormone receptor-positive
breast cancer with endocrine therapy [60]. Pathways associated with microbial infection,
which are Shigellosis or intestinal Shigella infection [61], bacterial invasion of epithelial cells,
pathogenic *Escherichia coli* infection, viral myocarditis and prion diseases, were enriched as
well.

Fc gamma receptor (FCGR) mediated phagocytosis is enriched in both KEGG and Reac-
tome analyses (Fig 7). Terms related to platelet activation and aggregation are over-repre-
sented in the Reactome database enrichment (Fig 7b). These include "Integrin alpha IIb/beta 3
signaling" [62], "Platelet activation, signaling and aggregation", "Platelet Aggregation (Plug
Formation)". Furthermore, involvement of the Pirin signaling in immune system pathways is
also highlighted via enrichment of "Chemokine signaling pathway", "Immune system",
"Innate immune system", "CD28 dependent Vav1 pathway" and "CD28 co-stimulation" [63].

**Enriched diseases.** Enrichment analysis was performed to identify the contribution or
possible association of Pirin and its interaction partners with diseases. Thirty nine of the
enriched diseases with adjusted p-value <0.05 were considered as significant and are shown in
Fig 8a. To further narrow down the exploration, a GDA network for Pirin, BCL3, NFIX,
NFKBIA, SMAD9 and RELA was obtained from Cytoscape based on curated data in DisGe-
NET (Fig 8b).

Liver carcinoma is the most significantly enriched disease for the Pirin interactome (Fig
8a). Both breast carcinoma and malignant neoplasm of breast are enriched for 30 of the query

**Table 3.** *PIR* gene variants with regulatory features.

| Variant | Consequence | Gene | Related regulatory elements | rVarBase identified target genes |
|---|---|---|---|---|
| rs73635099 | Upstream gene variant | PIR | TF binding region; Chromatin interactive region | PIR; ACE2; BMX |
| rs73635098 | Upstream gene variant | PIR | TF binding region; Chromatin interactive region | PIR; CA5BP1; BMX |
| rs574000198 | Upstream gene variant | PIR | TF binding region; Chromatin interactive region | PIR; CA5BP1; BMX; ACE2 |
| rs565388697 | Upstream gene variant | PIR | TF binding region; CpG island; Chromatin interactive region | PIR; CA5BP1;BMX |
| rs539142342 | Upstream gene variant | PIR | TF binding region; CpG island; Chromatin interactive region | PIR; BMX; CA5BP1 |
| rs4830964 | Upstream gene variant | PIR | TF binding region; Chromatin interactive region | PIR; BMX; CA5BP1; ACE2 |
| rs4830963 | Upstream gene variant | PIR | TF binding region; Chromatin interactive region | PIR; CA5BP1; BMX |
| rs376783522 | Upstream gene variant | PIR | TF binding region; Chromatin interactive region | PIR; CA5BP1; BMX |
| rs375074415 | Upstream gene variant | PIR | TF binding region; Chromatin interactive region | PIR; ACE2; BMX |
| rs372053429 | Upstream gene variant | PIR | TF binding region; Chromatin interactive region | PIR; ACE2; BMX |
| rs369918088 | Upstream gene variant | PIR | TF binding region; CpG island; Chromatin interactive region | PIR; BMX; CA5BP1 |
| rs191922328 | Upstream gene variant | PIR | TF binding region; Chromatin interactive region | PIR; BMX; CA5BP1 |
| rs190853164 | Upstream gene variant | PIR | TF binding region; Chromatin interactive region | PIR; BMX; ACE2 |
| rs189179622 | Upstream gene variant | PIR | TF binding region; Chromatin interactive region | PIR; ACE2; BMX |
| rs188792194 | Upstream gene variant | PIR | TF binding region; Chromatin interactive region | PIR; BMX; CA5BP1 |
| rs186401460 | Upstream gene variant | PIR | TF binding region; Chromatin interactive region | PIR; ACE2; BMX |
| rs186082169 | Upstream gene variant | PIR | TF binding region; Chromatin interactive region | PIR; ACE2; BMX |
| rs185489142 | Upstream gene variant | PIR | TF binding region; Chromatin interactive region | PIR; CA5BP1; BMX |
| rs181235643 | Upstream gene variant | PIR | TF binding region; Chromatin interactive region | PIR; ACE2; BMX |
| rs149701479 | Upstream gene variant | PIR | TF binding region; Chromatin interactive region | PIR; CA5BP1; BMX; ACE2 |
| rs147643730 | Upstream gene variant | PIR | TF binding region; Chromatin interactive region | PIR; BMX; ACE2 |
| rs145271657 | Upstream gene variant | PIR | TF binding region; Chromatin interactive region | PIR; ACE2; BMX |
| rs144528782 | Upstream gene variant | PIR | TF binding region; Chromatin interactive region | PIR; BMX; CA5BP1; ACE2 |
| rs140384063 | Upstream gene variant | PIR | TF binding region; Chromatin interactive region | PIR; CA5BP1; BMX |
| rs113506453 | Upstream gene variant | PIR | TF binding region; Chromatin interactive region | PIR; CA5BP1; BMX |
| rs908005 | Intron variant | PIR | TF binding region; Chromatin interactive region | PIR; BMX |
| rs76433744 | Intron variant | PIR | TF binding region | PIR |
| rs73635097 | Intron variant | PIR | TF binding region; Chromatin interactive region | PIR; BMX |
| rs73635096 | Intron variant | PIR | TF binding region; Chromatin interactive region | PIR; BMX |
| rs73449347 | Intron variant | PIR | TF binding region | PIR |
| rs6629105 | Intron variant | PIR | TF binding region; Chromatin interactive region | PIR; BMX |
| rs6629104 | Intron variant | PIR | TF binding region; Chromatin interactive region | PIR; BMX |
| rs60378841 | Intron variant | PIR | TF binding region | PIR |
| rs5980162 | Intron variant | PIR | TF binding region; Chromatin interactive region | PIR; BMX |
| rs58914845 | Intron variant | PIR | TF binding region | PIR |
| rs561645554 | Intron variant | PIR | TF binding region; Chromatin interactive region | PIR; BMX |
| rs560217510 | Intron variant | PIR | TF binding region | PIR |
| rs546798401 | Intron variant | PIR | TF binding region; Chromatin interactive region | PIR; BMX |
| rs528626194 | Intron variant | PIR | TF binding region; Chromatin interactive region | PIR; BMX |
| rs376208082 | Intron variant | PIR | TF binding region; Chromatin interactive region | PIR; CA5BP1; BMX |
| rs373370797 | Intron variant | PIR | TF binding region; Chromatin interactive region | PIR; BMX |
| rs371317022 | Intron variant | PIR | TF binding region | PIR |
| rs371266284 | Intron variant | PIR | TF binding region; Chromatin interactive region | PIR; BMX |
| rs368755567 | Intron variant | PIR | TF binding region; Chromatin interactive region | PIR; BMX |
| rs367898413 | Intron variant | PIR | TF binding region; Chromatin interactive region | PIR; BMX |
| rs35612843 | Intron variant | PIR | TF binding region | PIR |
| rs34107232 | Intron variant | PIR | TF binding region | PIR |

*(Continued)*

**Table 3.** (Continued)

| Variant | Consequence | Gene | Related regulatory elements | rVarBase identified target genes |
|---|---|---|---|---|
| rs2095 | Intron variant | PIR | TF binding region; Chromatin interactive region | PIR; BMX; CA5BP1 |
| rs2094 | Intron variant | PIR | TF binding region; Chromatin interactive region | PIR; BMX |
| rs201190104 | Intron variant | PIR | TF binding region | PIR |
| rs193189971 | Intron variant | PIR | TF binding region | PIR |
| rs190912217 | Intron variant | PIR | TF binding region; CpG island; Chromatin interactive region | PIR; BMX |
| rs190521888 | Intron variant | PIR | TF binding region; Chromatin interactive region | PIR; BMX |
| rs190208998 | Intron variant | PIR | TF binding region; Chromatin interactive region | PIR; BMX |
| rs188344505 | Intron variant | PIR | TF binding region | PIR |
| rs187777697 | Intron variant | PIR | TF binding region; CpG island; Chromatin interactive region | PIR; BMX |
| rs187257498 | Intron variant | PIR | TF binding region | PIR |
| rs185541409 | Intron variant | PIR | TF binding region; Chromatin interactive region | PIR; BMX |
| rs184665224 | Intron variant | PIR | TF binding region | PIR |
| rs184237791 | Intron variant | PIR | TF binding region | PIR |
| rs182838318 | Intron variant | PIR | TF binding region; Chromatin interactive region | PIR; BMX |
| rs181606495 | Intron variant | PIR | TF binding region; Chromatin interactive region | PIR; BMX |
| rs16979911 | Intron variant | PIR | TF binding region | PIR |
| rs16979910 | Intron variant | PIR | TF binding region | PIR |
| rs1567894 | Intron variant | PIR | TF binding region; CpG island; Chromatin interactive region | PIR; BMX |
| rs149438787 | Intron variant | PIR | TF binding region; Chromatin interactive region | PIR; BMX |
| rs12852159 | Intron variant | PIR | TF binding region; Chromatin interactive region | PIR; BMX |
| rs12851908 | Intron variant | PIR | TF binding region; Chromatin interactive region | PIR; BMX |
| rs12850489 | Intron variant | PIR | TF binding region; Chromatin interactive region | PIR; BMX |
| rs12560046 | Intron variant | PIR | TF binding region; Chromatin interactive region | PIR; BMX |
| rs112327224 | Intron variant | PIR | TF binding region | PIR |
| rs111378408 | Intron variant | PIR | TF binding region; Chromatin interactive region | PIR; BMX |
| rs111284672 | Intron variant | PIR | TF binding region; Chromatin interactive region | PIR; BMX |
| rs372563960 | 5' UTR variant | PIR | TF binding region; CpG island; Chromatin interactive region | PIR; CA5BP1; BMX |
| rs2271550 | 5' UTR variant | PIR | TF binding region; Chromatin interactive region | PIR; CA5BP1; BMX |
| rs183910461 | 5' UTR variant | PIR | TF binding region; CpG island; Chromatin interactive region | PIR; CA5BP1; BMX |
| rs148530113 | 5' UTR variant | PIR | TF binding region; CpG island; Chromatin interactive region | PIR; BMX; CA5BP1 |
| rs180708754 | 5' UTR variant | PIR | miRNA target site | PIR |

genes. Other over-represented malignancies for the query genes include malignant neoplasm of mouth, malignant neoplasm of prostate, lip and oral cavity carcinoma, secondary malignant neoplasm of lymph node, and prostate carcinoma. Multiple neoplasms (benign or malignant tumor) are correlated to RELA, NFKBIA and SMAD9 (Fig 8b).

The second most significantly enriched disease is Alzheimer's disease (AD) (Fig 8a). Neuroinflammation contributes to onset and progression of this neurodegenerative disorder [64, 65]. Along with Alzheimer's disease, multiple other degenerative brain disorders are correlated to at least 3 of the query genes. The broader umbrella term motor neuron disease (MND) or atrophy as well as a number of neurodegenerative conditions described by the terms, including spinal muscular atrophy and the Guam form of amyotrophic lateral sclerosis (ALS), are enriched as well [66–68]. Several forms of Parkinsonian disorder or parkinsonism are also enriched significantly. Such neurodegenerative disorders are characterized by rigidity, bradykinesia, resting tremor, and postural instability [69–71] and the major hallmark of Parkinson's disease (PD) is chronic inflammation [72].

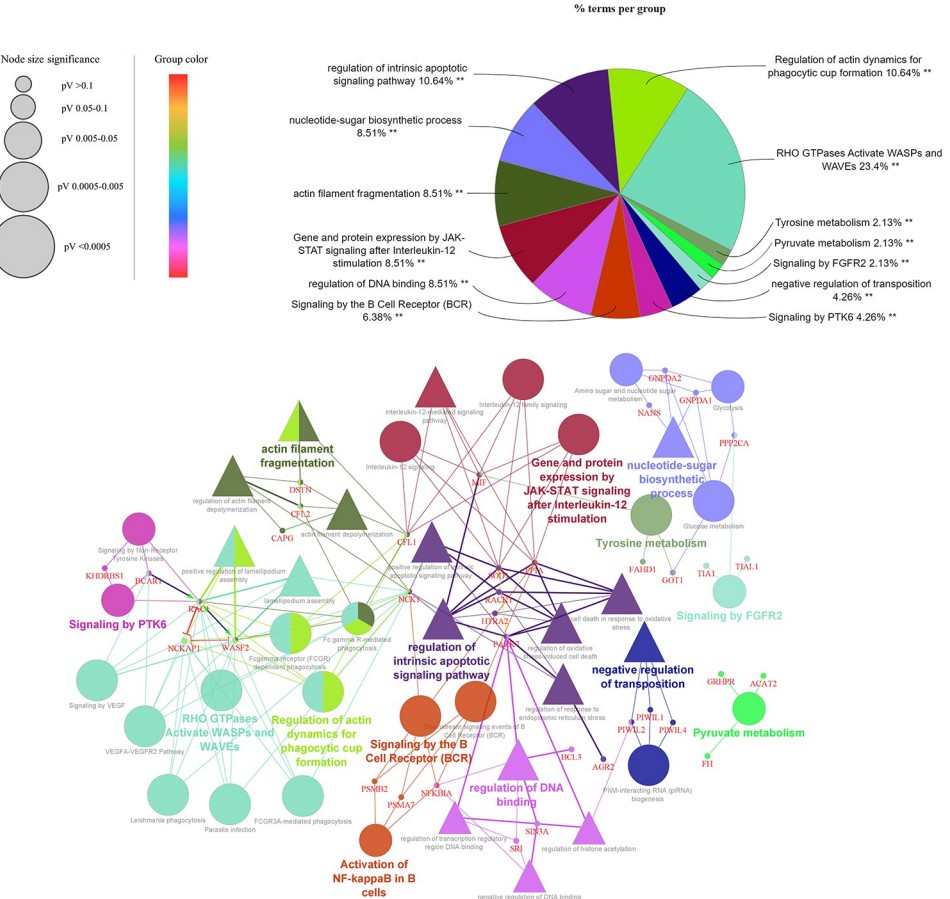

**Fig 6. ClueGO generated functional network of Pirin.** GO:BP terms and pathway terms are represented by triangles and ellipses, respectively. Interactors of Pirin are linked to enriched biological processes and KEGG and REACTOME pathways. Overrepresented functional categories are shown in the pie chart where the asterisks indicate significance of the terms (** p- value < 0.001).

A transient ischemic attack (TIA), that frequently precedes ischemic stroke, is an episode of neurological dysfunction resulting from inadequate blood flow to focal brain, spinal cord, or retina, without acute infarction [73, 74]. Multiple forms of transient neurological dysfunctions are highly clustered for the Pirin interactome (Fig 8a). Furthermore, Ramsay Hunt Paralysis Syndrome, a neurological disorder caused by varicella-zoster virus reactivation and replication at the facial nerve, correlates to interaction partners of Pirin. These enrichments indicate towards involvement of Pirin signaling in neuroinflammation and pathological neuro-alterations.

Not many diseases were connected to Pirin directly (Fig 8b). One of the reasons may be a lack of studies on this protein. Skin and connective tissue diseases are connected to Pirin, RELA and NFKBIA. RELA is also associated with endocrine system and metabolic diseases, including non-insulin dependent diabetes mellitus, diabetic angiopathies and diabetic nephropathy as well as digestive system diseases, such as biliary cirrhosis, hepatomegaly, hepatic insufficiency, experimental and alcoholic liver cirrhosis, ulcerative colitis and cholestasis. SMAD9 is correlated to various respiratory tract diseases and different types of pulmonary hypertension.

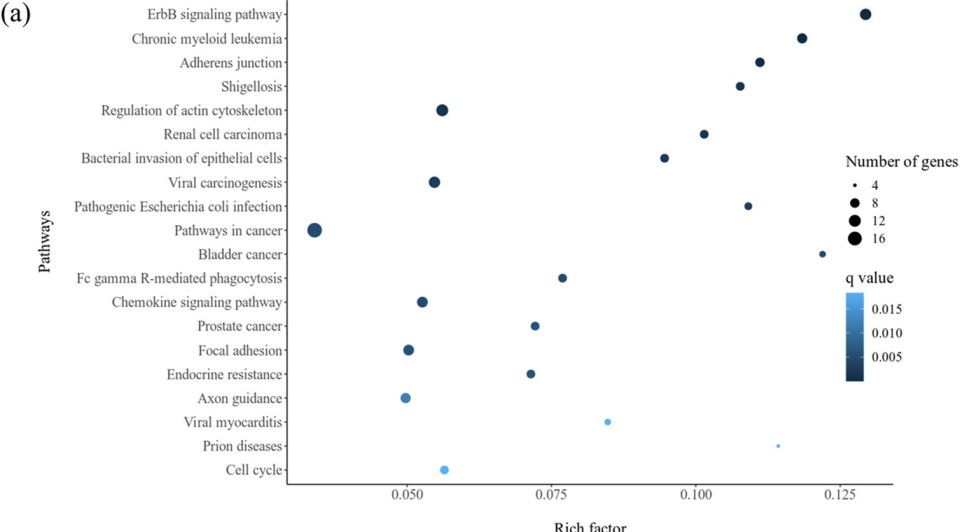

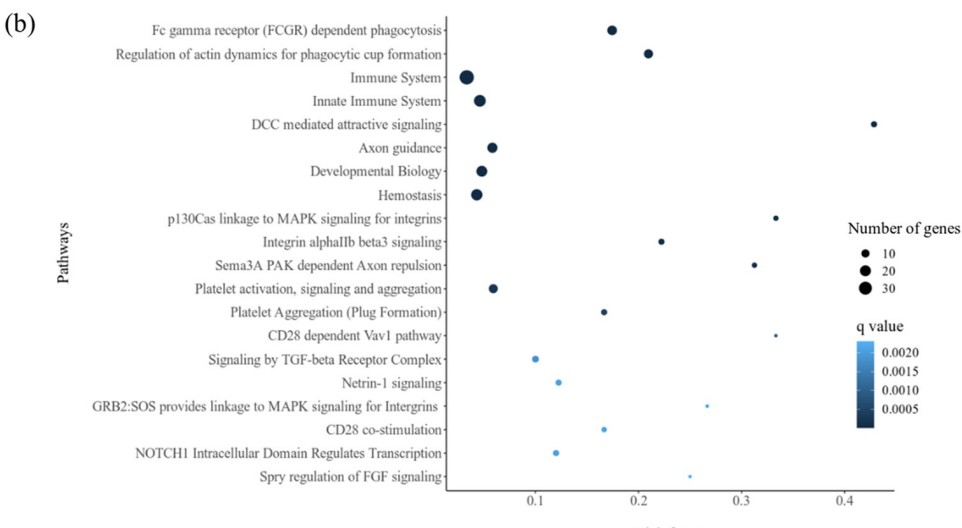

**Fig 7. Scatter plots for top 20 enriched pathways based on KEGG (A) and Reactome (B) pathway databases.** The x-axis represents the rich factor of each pathway and y-axis shows the pathways. The Rich factor refers to the ratio of number of target genes (nodes of the minimum network) annotated in a pathway term to all gene numbers annotated in that pathway term. The larger the Rich factor, the higher the degree of pathway enrichment. The size and color of the dots represent the number of candidate genes mapped to the indicated pathway and q value, respectively. A lower q value, that is the corrected p-value ranging from 0 ~ 1, indicates greater pathway enrichment.

## Regulatory variants of *PIR* gene

Using the rVarBase [52] database, 78 variants potentially modifying *PIR* gene expression were identified (Table 3). Among these, five, twenty five and forty eight are located in the 5' UTR, upstream and intron regions of *PIR* gene, respectively. Genetic variants residing in noncoding regions can be significantly associated with expression of corresponding genes [75]. The target genes for majority of the intron variants are *PIR* and *BMX* non-receptor tyrosine kinase (BMX).

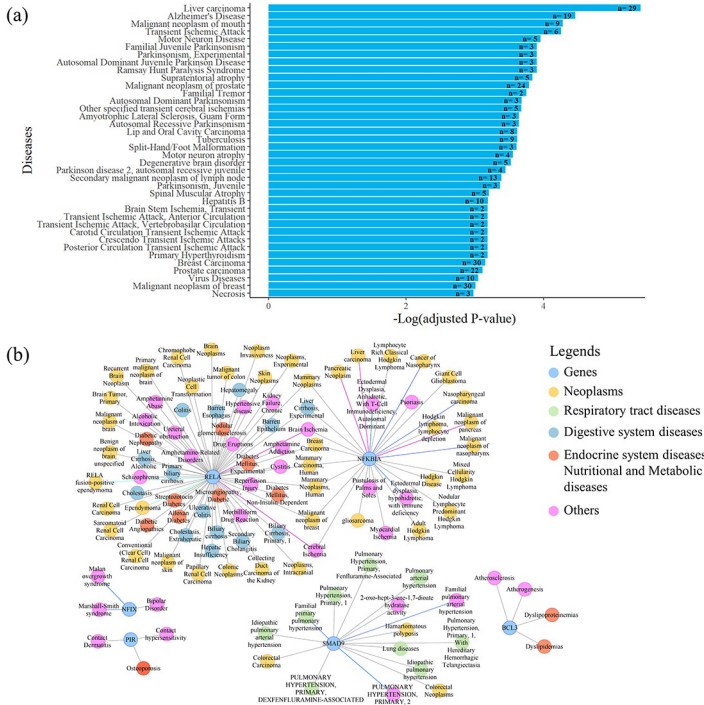

**Fig 8. Diseases significantly enriched for Pirin and its interaction partners.** (a) Diseases over-represented for minimum network by DisGeNET at EnrichR reflecting all interactor connected signaling of Pirin. The x-axis shows the significantly enriched disease terms and the y-axis represents the negative log (base 10) of the adjusted p-value. The number of genes (n) assigned to the disease term are shown with each bar. (b) GDA network with score 0 ~ 1 for Pirin and proteins that directly interact with it. Nodes are colored according to disease classes which are indicated on the right side.

rs180708754, a 3' UTR variant of *PIR* gene, is located within the target site of a miRNA (Table 3). According to mirSNP database [76], this SNP can break the binding of hsa-miR-3673 and hsa-miR-4694-3p whereas it enhances the binding of hsa-miR-5680 (data not shown). TargetScan (Release 7.1) [77] predicted *PIR* to be one of the target genes for both miR-4694-3p and hsa-miR-5680.

The twenty five upstream variants of *PIR* gene are overlapped by both Transcription factor (TF) binding region and chromatin interactive region (Table 3). Variants of TF binding region may affect gene expression by altering binding of TFs, which are molecular switches regulating the amount and timing of gene transcription via sequence-specific binding [75]. As distal regulatory elements interact with their target genes through long-range chromatin loop bridges [78, 79], variants within chromatin interactive regions can plausibly influence gene expression through deviating from the natural interaction. The potential target genes for the upstream regulatory variants of *PIR* are *PIR* itself, *ACE2*, *BMX*, and Carbonic Anhydrase 5B Pseudogene 1 (*CA5BP1*).

Nine variants (rs565388697, rs539142342, rs369918088, rs190912217, rs187777697, rs1567894, rs372563960, rs183910461, and rs148530113) reside on a CpG island (Table 3). Discrete CpG dinucleotide motifs are the sites for DNA methylation- a type of epigenetic modification, and genetic variants located within or close to high density CpG sites can be mediators of gene expression [80, 81].

### eQTLs in *PIR* and their geographic distribution

The GTEx project provides a comprehensive picture of expression quantitative trait loci (eQTL) that can alter gene expression profiles in a number of tissues by assessing the effects of genetic variations on transcriptomes across > 50 tissues of 838 individuals [54]. Among the identified regulatory *PIR* gene variants, nine (rs4830964, rs2094, rs2095, rs908005, rs1567894, rs2271550, rs5980162, rs6629104 and rs6629105) could be identified through GTEx portal as modulators of expression of *PIR* and Vascular Endothelial Growth Factor D (*VEGFD*) genes across multiple tissues (Table 4). rs6629104 and rs6629105 can affect expression of both *ACE2* and *PIR* in nucleus accumbens of basal ganglia within the brain (S2 Table). rs6629105 T allele increases expression of *ACE2* in nucleus accumbens (S1 Fig). rs4830964, rs2094, rs1567894, and rs6629104 regulates expression of carbonic anhydrase 5B pseudogene 1 (*CA5BP1*) in the lung. Variant allele frequencies (VAFs) at these loci in five super-populations (African, Admixed American, East Asian, European and South Asian) are given in Table 3. The frequencies of variant alleles at rs4830964, rs5980162, rs2095, rs2094 and rs1567894 are relatively low in South Asian populations compared to the other populations. The variant allele at rs2271550 is present with > 0.3 frequency only in the European population.

## Discussion

Persistent cellular stress induced perpetuation and uncontrolled amplification of inflammatory response results in a shift from tissue repair toward collateral damage, significant alterations of tissue functions, and derangements of homeostasis which in turn can lead to a large number of acute and chronic pathological conditions, such as chronic heart failure, atherosclerosis, myocardial infarction, neurodegenerative diseases, diabetes, rheumatoid arthritis, and cancer [82]. Keeping the vital role of balanced inflammation in maintaining tissue integrity in mind, the way to combating inflammatory diseases may be through identification and characterization of mediators of inflammation that can be targeted without hampering normal body function. In our study, we have demonstrated Pirin protein to have role in multiple important processes including inflammation, rearrangement of actin cytoskeleton as well as metastasis, ischemic attack and neurodegenerative diseases among others. We have analyzed regulatory variants of Pirin gene to have insight into its expression.

**Table 4. GTEx validated eQTLs and their frequencies in five super-populations.**

| eQTLs | GTEx validated target genes | Reference allele | Variant allele | VAFs | | | | | |
|-------|------------------------------|------------------|----------------|------|-----|-----|-----|-----|-----|
| | | | | All* | AFR | AMR | EAS | EUR | SAS |
| rs4830964 | PIR; VEGFD; CA5BP1; FANCB; TMEM27 | C | G | 0.480 | **0.803** | **0.452** | **0.420** | **0.386** | 0.214 |
| rs113506453 | CA5BP1 | A | G | 0.038 | 0.129 | 0.029 | 0 | 0 | 0 |
| rs908005 | PIR; VEGFD; TMEM27; CA5BP1 | C | T | 0.291 | **0.386** | 0.267 | 0.017 | **0.354** | **0.403** |
| **rs6629105** | PIR; VEGFD; TMEM27; CA5B; CA5BP1; **ACE2** | G | T | **0.333** | 0.151 | 0.323 | **0.551** | 0.253 | **0.446** |
| **rs6629104** | PIR; VEGFD; TMEM27; CA5B; CA5BP1; **ACE2** | T | C | **0.343** | 0.157 | 0.328 | **0.558** | 0.264 | **0.468** |
| rs5980162 | PIR; VEGFD; TMEM27 | G | A | **0.353** | **0.413** | **0.397** | **0.425** | **0.383** | 0.130 |
| rs2095 | PIR; VEGFD; TMEM27 | C | T | **0.320** | 0.293 | **0.386** | **0.425** | **0.384** | 0.130 |
| rs2094 | PIR; VEGFD; CA5BP1; FANCB; TMEM27 | C | T | **0.479** | **0.798** | **0.452** | **0.425** | **0.385** | 0.212 |
| rs1567894 | PIR; VEGFD; CA5BP1; FANCB | C | T | 0.466 | **0.751** | **0.445** | **0.425** | **0.385** | 0.212 |
| rs2271550 | PIR; VEGFD; TMEM27; CA5B; CA5BP1 | C | A | 0.163 | 0.010 | 0.216 | 0.018 | **0.351** | 0.291 |

*All; AFR- African; AMR- Ad Mixed American; EAS- East Asian; EUR- European; SAS- South Asian

## Mode of interaction of Pirin with its direct interaction partners and associated consequences

The four proteins that directly interact with Pirin are NFIX, BCL3, NFKBIA and SMAD9 (Table 1). Pirin was identified through Y2H system based screening of the interaction partners of NFIX [6, 83]. NFIX is a master regulator for metastasis of lung cancer [84]. The iron cofactor is required for interaction of Pirin with NFIX and BCL3 [8]. The $Fe^{2+}$ bound conformation of Pirin appears to bind to the putative binding regions of the direct interactors with higher affinities and has more resemblance to the native pose in comparison with the $Fe^{3+}$ bound conformation (Figs 1–4 and Table 2).

**Regulation of NF-κB signaling pathway.** The NF-κB family of transcription factors (TFs) consists of five members- p65 (RelA), RelB, c-Rel, p50 (NF-κB1), and p52 (NF-κB2), which form transcriptionally active homodimers (except Rel B) and heterodimers [5, 85, 86]. In unstimulated cells, NF-κB dimers are sequestered in an inactive form through association with one of three typical inhibitor of NF-κB (IκB) proteins: IκBα (encoded by NFKBIA), IκBβ (NFKBIB), and IκBε (NFKBIE), or the precursor proteins p100 (NFKB2) and p105 (NFKB1) or the atypical IκB proteins: IκBζ, BCL3 and IκBNS via the ankyrin repeat domains of these inhibitors [85, 87, 88]. Upon stimulation of cells by agents, such as TNFA, IL1 or numerous pathogens or pathogenic proteins, like members of the *Poxviridae*, N-protein of SARS-CoV-2 as well as pathogenic bacteria, the IκB kinase (IKK) complex is activated that in turn phosphorylates IκB molecules on two serine residues and mediates their ubiquitination by the SCF E3 ligase and subsequent degradation by the 26S proteasome [4, 87, 89, 90]. As a result of degradation of the IκB inhibitors, NF-κB dimer is freed and able to translocate into the nucleus to initiate a transcriptional response [4].

Various NF-κB complexes, predominantly the p65/p50 heterodimers activated via IκBα-degradation regulate transcription of target genes in the canonical pathway, while the RelB/p52 heterodimer formed via inducible proteasomal processing of p100 to p52 drives a transcriptional response in the non-canonical pathway [88, 91]. Sequestration by IκBα is important for modulating NF-κB signaling as functional polymorphisms in the NFKBIA promoter and 3′ untranslated regions are significantly associated with cancers, such as HBV-induced hepatocarcinogenesis [92, 93]. Co-fractionation process indicated interaction (indirect) between Pirin and ubiquitin-conjugating enzyme E2 L3 (UBE2L3) (Table 1), which is a ubiquitin-conjugating enzyme with roles in ubiquitination cascade of proteins for generating signals for 26S proteasome dependent protein degradation (KEGG: hsa04120) [45]. Pirin signaling includes two subunits of 26S proteasome that are proteasome subunit alpha type-7 (PSMA7) and proteasome subunit, beta type, 2 (PSMB2) [94], among which PSMB2 is a first degree interactor of Pirin (Table 1 and Fig 5). Pirin may contribute to NF-κB signaling through direct interaction with IκBα (*NFKBIA*) and subsequently mediating its proteasomal degradation. The nuclear localization of Pirin contradicts such role, but presence of Pirin in cytoplasm in a subset of melanomas has been reported previously [7]. Additionally, despite an exclusively cytosolic steady state localization of IkBα/p50 complexes, these are constantly shuttled between the cytosol and nucleus using the nuclear export sequence (NES) of IkBα and the nuclear localization sequence (NLS) of p50 [5]. Thus, it might not be unlikely that over-expression of Pirin dysregulate sequestration of NF-κB by IκBα and lead to associated diseases.

The carboxy-terminal transactivation domains, that activate transcription at the κB-sites in target genes, are present in RelA, RelB and c-Rel, but absent in the p50 and p52 homodimers [86]. BCL3 is a transactivation domain containing oncoprotein that is located predominantly in the nucleus [95]. BCL3 provides transactivation domains to p50 and p52 homodimers by interacting with these complexes and promote transcription of NF-κB target genes [96].

Without any direct association with p50, Pirin enhances the amount of the DNA binding activity of p50-BCL3 by interacting with the ankyrin repeat domains of BCL3 and forming a quaternary complex [8, 97].

Along with activation, binding of BCL3 to p50 and p52 homodimers can lead to repression of a subset of NF-κB regulated genes via stabilizing repressive p50 homodimers or recruiting co-repressors [96]. However, BCL3 can also dissociate repressive p50 homodimers from κB-sites on DNA by sequestering these in a complex, which allows p65/p50 heterodimers to activate transcription at these sites and mediate canonical signaling pathway [96, 97]. Such removal of p50 and p52 homodimers from bound DNA is regulated by unphosphorylated BCL3 that plays inhibitory roles similar to a typical IκB family member [98]. One of the first degree interactors of Pirin is PPP2CA (Fig 5), which is a serine/threonine phosphatase with activities critical for maintaining healthy cellular functions and suppressing tumor [99]. Pirin may interact with BCL3 and PPP2CA simultaneously to mediate the shift from non-canonical to canonical NF-κB signaling through dephosphorylation of BCL3, although still there is no study confirming any such role.

**Shifting between different modes of NF-κB signaling pathway.** Change in the oxidation state causes structural change in Pirin through altering a R-shaped surface loop that includes the area surrounding the N-terminal metal-binding cavity and the interface between the two cupin domains [9]. This R-shaped surface region interacts with the C-terminal Rel homology domain of p65 and the $Fe^{3+}$ conformation enhances the binding of p65 to the κB-target site [9]. As the iron cofactor plays an important role in interaction between BCL3 and Pirin [8], change in the distance of the metal ligand from the surface by change in oxidation state [9] can be the reason why BCL3 does not interact with same energy with both the conformations of Pirin (Fig 1 and Table 2). It appears that the Fe(II) conformation mediates the non-canonical NF-κB through strengthening the function of p50-BCL3 on κB-sites and the Fe(III) conformation with no such role on p50-BCL3-DNA complex contributes critically to the p65 regulated canonical pathway with a key role of redox state of cell. This may explain the shift between different NF-κB pathways with full reversibility [9].

**Transcriptional co-regulation.** BCL3 expression is elevated in various hematopoietic and solid cancers, including breast and hepatocellular carcinomas [95, 100]. This oncoprotein acts as a bridging factor between NF-κB and nuclear co-regulators [97]. The same conformation, but completely different regions (as observed by overlaying the docked structures in Chimera 1.15rc [101], of Pirin interacts with NFIX and BCL3 (Figs 1 and 3). NFI family members can modulate activation and repression of transcription of genes [6]. Pirin may connect NFIX to BCL3-p50 or BCL3-p52 homodimer bound genes, but definitive conclusions regarding effect of NFIX on transcription of κB-genes await further study. Based on the *in silico* data (Table 1 and Fig 4), SMAD9 directly interacts with Pirin. SMAD9 is a transcriptional repressor of bone morphogenetic protein (BMP) signaling [102]. Rare sequence variants in SMAD9 were reported to contribute to pathogenesis of pulmonary arterial hypertension [103]. SMAD9 protein is associated with colorectal neoplasms, familial pulmonary hypertension and pulmonary arterial hypertension (Fig 7).

## Functional and disease enrichment for Pirin signaling

**Actin cytoskeleton remodeling and regulation of metastasis of cancer cells.** Pirin signaling is significantly involved in regulation of actin dynamics, actin filament fragmentation as well as Rho GTPases activated WASPs and WAVEs pathway (Figs 6 and 7). Through activating Wiskott–Aldrich syndrome family of proteins, such as WASP and WAVE1/2, Rho GTPases stimulate formation of lamellipodia and filopodia that are involved in directional

motility of cells and invasiveness and metastasis of cancer cells [104]. Pirin signaling includes (Table 1) Wiskott-Aldrich syndrome protein family member 2 (WASF2) and RAC1 which are associated with lamellipodia assembly (Fig 5). Pirin also interacts with cytoplasmic protein NCK1, which mediates cancer metastasis [105, 106] as well as functions as an intracellular messenger leading to angiogenesis in the ERBB signaling pathway (KEGG entry: hsa04012) [45]. RAC1, together with NCK, mediates activation of the Arp2/3 complex leading to polymerization of the actin cytoskeleton to form lamellipodia typical of mesenchymal movements as well as tumor metastasis and angiogenesis [107, 108]. So, role of Pirin in cancers appears to be more centered to migration and invasion of tumor cells, instead of growth and initiation.

The most enriched KEGG [45] pathways and DisGeNET [51] analyzed diseases for interactors of Pirin are associated with cancers (Figs 7 and 8). Earlier reports also suggest a role Pirin in different cancers [16, 17, 109, 110]. Interrupting the interaction between Pirin and BCL3 with a Pirin inhibitor has been shown to inhibit melanoma cell metastasis via suppression of snail homolog 2 (SNAI2) [16]. Pirin can also mediate metastasis of cervical cancer cells independent of BCL3-SNAI2 signaling [17]. In oral and cervical cells, *PIR* gene silencing with small interfering RNA (siRNA) was shown to increase E-cadherin transcripts as well as reduce Vimentin, Slug, Zeb and Snail transcripts in oral and cervical cancer cells [109, 110]. Furthermore, high-risk human papillomavirus (HR-HPV) oncoproteins enhance levels of Pirin in both epithelial cervical and oral cancer cells [109]. In oral tumor cells, HPV16 E7 oncoprotein mediated induction of the EGFR/PI3K/AKT1/NRF2 pathway and recruitment of NRF2 in the *PIR* promoter was found to lead to Pirin/NF-κB activation which in turn enhanced epithelial to mesenchymal transition (EMT) and cell migration [10, 111].

Pirin is involved in a pro-fibrotic signaling pathway depending on myocardin-related transcription factor (MRTF) and serum response factor (SRF) which are activated downstream of the Rho GTPases [112]. Along with these enhancers of metastasis, Pirin also interacts with destrin (DSTN), cofilin (CFL) and gelsolin-like actin-capping protein CAPG (Table 1), which have opposite effects and play a role in actin filament fragmentation/depolymerization (Fig 6). Cofilin reverses the process of polymerization by converting F-actin filaments into G-actin monomers, but this reversal can be inactivated by RAC1 [107]. Rho GDP dissociation inhibitor (GDI) alpha (ARHGDIA) has a first degree association with Pirin (Fig 5). ARHGDIA is an ubiquitously expressed interactor of several Rho GTPases, including RAC1 and it blocks activation of Rho proteins through sequestration of the inactive GDP-bound Rho proteins in the cytosol and inhibiting the switch to active GTP-bound states [113]. Association between Pirin and ARHGDIA may impede this blockage and allow Rho GTPases to exert their function. Pirin also interacts with breast cancer anti-estrogen resistance 1 (BCAR1) protein that activates RAC1 (Fig 5). The exact mechanism of how interaction with these proteins contribute to Pirin mediated metastasis of cancer cells is yet to be known, but Pirin signaling appears to play a key role in cancer invasiveness and migration.

**Role in neuropathological complications.** Various neurodegenerative conditions, including Alzheimer's disease, motor neuron disease, Parkinsonism, Ramsay Hunt paralysis syndrome and amyotrophic lateral sclerosis (Guam form) are enriched for Pirin and its partnering proteins (Fig 8a). Abundant abnormal aggregates of neuronal cytoskeletal proteins are signatures of many neurodegenerative diseases [114]. Balanced cofilin activity is a prerequisite for actin turnover and proper central nervous system (CNS) functions [115].

Piwi-like proteins, including as PIWIL1, PIWIL2, and PIWIL4, are among the Pirin interactors (Table 1). These proteins are associated with negative regulation of transposition (Fig 6). These proteins play a vital role in maintaining piRNA activity, which is increased in neuropathological conditions [116]. In previous studies, PIWIL1, PIWIL2, and PIWIL4 have been reported to function in cancer cell proliferation, metastasis, and invasion [117, 118].

**Involvement in infection associated body processes and complications.** For Pirin and its interaction partners, disease conditions correlated to pathogenic infections, such as Tuberculosis, Hepatitis B and viral diseases, as well as the body's response to the infections, including regulation of actin dynamics for phagocytic cup formation and Fc gamma receptor (FCGR) dependent phagocytosis, are enriched (Figs 6 and 8). Phagocytic cups, an actin-based structure at the plasma membrane of a phagocyte, are essential for phagocytosis and digestion of pathogens, and WASP plays a key role in their formation [119, 120]. In addition, activation of RAC1 and inhibition of cofilin is part of the regulation of actin cytoskeleton in FCGR mediated phagocytosis (KEGG entry: hsa04666) [45], which may be mediated by Pirin, as mentioned previously. Furthermore, the relation of viral infection to the development of neurological disorders such as Parkinson's disease, Alzheimer's disease, and multiple sclerosis is well known [121]. For example, SARS-CoV-2 can trigger cellular processes involved in acute and subacute neurodegeneration by entering the brain [122, 123]. Involvement of Pirin signaling in actin cytoskeleton reorganization, phagocytic cup formation, and interaction with actin filament modulating proteins suggest association of excessive Pirin production with the development of neurological complications, in presence or absence of infection.

**Oxidative stress induced regulation of cell death.** Basal expression of *PIR* can be modulated by nuclear factor (erythroid-derived 2)-like 2 (NRF2; encoded by the *NFE2L2* gene) transcription factor through functional antioxidant response elements (AREs) in its promoter region [124, 125]. Knockdown of NRF2 in HeLa cells resulted in decreased PIR mRNA and protein levels [126]. NRF2 is triggered in mammals as a protective response to cellular oxidative, inflammatory and electrophilic stress [124, 127] may lead to excessive Pirin production.

Regulation of intrinsic apoptotic signaling pathway, oxidative stress induced cell death and response to endoplasmic reticulum stress are significantly enriched for Pirin interactome and interactors. For example, NCK1, superoxide dismutase 1 (SOD1), Parkinson protein 7 (PARK7), peptidylprolyl isomerase A (PPIA), receptor for activated C kinase (RACK1), and HtrA serine peptidase 2 (HTRA2) are associated with these processes (Fig 6). Upregulation of Pirin in the airway epithelium in response to the acute oxidative stress imposed by cigarette smoke was found to induce apoptosis [128]. Additionally, Pirin has been shown to play role in resistance to ferroptosis, which is an iron-dependent non-apoptotic cell death, in human pancreatic cancer cells [129]. On the other hand, Pirin was reported to be a negative regulator of senescence in melanocytic cells [130]. Thus, Pirin interactome is involved in regulation of apoptotic or non-apoptotic cell death but the mode of regulation cannot be inferred conclusively.

**Role in diabetic complications.** Oxidative stress has a major contribution to the pathogenesis both microvascular and cardiovascular diabetic complications [131]. As mentioned earlier, ferric conformation of Pirin plays a vital role in NF-κβ p65 pathway activation. The canonical NF-κB pathway mediated by P65/p50 heterodimer directly induces the production of proinflammatory cytokines, such as TNFA and IL6 [88]. Proinflammatory NF-κβ pathway plays a critical role in pathophysiology of diabetes as well as associated vascular complications, such as diabetic retinopathy, diabetic nephropathy and cardiomyopathy [132, 133]. Pirin, RELA (p65) and NFKBIA are significantly associated with endocrine system diseases (Fig 8b). Pirin hyperactivated by oxidative stress may contribute to pathological inflammation involved in diabetic complications.

**Involvement other conditions.** Transient ischemic attacks (TIAs) were enriched for Pirin signaling (Fig 8a). Recruitment of platelets to the ischemic region and their aggregation can result in TIAs [134, 135]. Reactome pathway enrichment analysis (Fig 7b) demonstrated involvement of Pirin interactome in platelet activation and aggregation as well as αIIb/β3 receptor signaling that is essential for platelet aggregation [62].

Additionally, glucose, amino acid and nucleotide sugar metabolism is significantly associated with Pirin interactome (Fig 6). Pirin is also associated to osteoporosis (Fig 8b) and NF-κB contributes to bone formation impairment in osteoporosis [136].

It appears that Pirin is associated with a myriad of inflammatory and neurodegenerative diseases as well as metastasis of cells through interaction with other proteins.

## Abnormal expression of Pirin by regulatory variants

Pirin production can be modulated by expression modifying (modifier) regulatory variants (Tables 3 and 4). CpG island downstream from the transcription start site (TSS) of the *PIR* gene is crucial for its expression [126]. For regulatory variants rs565388697, rs539142342, rs369918088, rs190912217, rs187777697, rs1567894, rs372563960, rs183910461, and rs148530113, the CpG island was found to be the related regulatory element (Table 3). rs180708754 can influence the binding of miRNA to the *PIR* gene.

*PIR* gene upstream variant rs4830964 and intron variant rs2094 may regulate expression of not only *PIR* gene, but also vascular endothelial growth factor D (*VEGFD*), carbonic anhydrase 5B pseudogene 1 (*CA5BP1*), Fanconi anemia group B protein (*FANCB*) and Transmembrane protein 27 (*TMEM27*) (Tables 2 and 3). Related regulatory elements for these two variants are TF binding region and chromatin interactive region. Both rs5980162 and rs1567894 are intron variants that regulate expression of *PIR* and *VEGFD* across various tissues (S2 Table). Variant alleles at rs4830964, rs2094, rs5980162 and rs1567894 exist with > 0.3 frequency in African, admixed American, East Asian and European populations but with < 0.3 frequency in South Asian populations (Table 4). This may cause disparity in *PIR* expression and lead to differential prevalence of Pirin mediated inflammatory conditions and aberrant actin cytoskeleton remodeling involved diseases (as discussed previously) among these populations. For example, NF-κB pathway, that is regulated by Pirin [9, 13], plays a central role in pro-inflammatory cytokine response observed in COVID-19 patients [137] and interestingly, COVID-19 death rate have been comparatively lower in the South Asian countries compared to the other nations in Europe, United states and Asia [138].

Expression of both *PIR* and *ACE2* can be regulated by rs6629104 and rs6629105 in nucleus accumbens of basal ganglia within the brain (Table 2 and S2 Table). mRNA abundance is an associated trait for these two intron variants of *PIR* (Table 2). Dysregulated *PIR* expression in brain regions may be associated with neurodegenerative diseases (as discussed above) as well as with neurological tumors [10]. Variant alleles at rs6629104 and rs6629105 are present with > 0.4 frequency in South- and East Asian populations, but with comparatively lower frequencies in African and European populations. ACE2 expression is enhanced by rs6629105 T allele in Nucleus accumbens (S1 Fig). Increased ACE2 expression in brain regions can be correlated to enhanced neuro-invasion of SARS-CoV-2 and associated neurological symptoms [139].

This study provides mechanistic insights into how Pirin's conformational alterations regulate its interaction with key inflammatory regulators, as well as its function in tumor invasion, and other metabolic and neuropathological complications. Further *in vivo* and *in vitro* studies will shed more light on such interactions. Because of its important role in critical pathways that are associated with multiple disease conditions, Pirin might be considered as an important drug target, particularly as a bypass route to suppress over-activation of the NF-κB pathway for treating chronic inflammation-mediated diseases. As the frequency of certain regulatory variants of the PIR gene varies among populations of different ethnicities, additional research on the variants of Pirin and associated phenotypic changes is necessary to determine their connection with disease conditions. These variants may be taken into consideration while

developing drugs or predicting outcomes. These PIR variants may also be taken into consideration while explaining the global incidences of associated disease conditions.

## Conclusion

Dysregulated inflammatory networks stand at the forefront of highly prevalent human diseases. Identification of novel inhibitory target(s) for pathogenic inflammation is highly needed. Due to the lack of studies on the functions of the Pirin protein, an interactome-based approach was applied for the characterization of its functionality. Both conformations, Fe(II)- and Fe(III)-bound, of this protein are intricately associated with the NF-κB inflammatory pathway. Along with its role in the regulation of NF-κB activity, Pirin signaling appears to play a role in cytoskeleton remodeling, metastasis and invasion of tumors, platelet aggregation, and stroke. Pirin may be an effective target for anti-inflammatory and anti-metastatic treatments. There are regulatory variants in the *PIR* gene that not only regulate the expression of *PIR* but also the *ACE2* gene, which is the targeted binding site of SARS-CoV-2.

## Supporting information

**S1 Table. Domains of binding partners of Pirin.**
(DOCX)

**S2 Table. eQTLs in PIR and their regulated genes across different tissues.**
(DOCX)

**S1 Fig. Violin plot representing normalized expression of PIR and ACE2 genes in nucleus accumbens region of brain depending on the genotype at rs6629105.** The number of subjects is shown under each genotype. The median value of the gene expression at each genotype is indicated by the white lines in the black box plots.
(PNG)

## Acknowledgments

The authors are thankful to the Ministry of Science and Technology, Bangladesh, for its support.

## Author Contributions

**Conceptualization:** Tamim Ahsan, Sabrina Samad Shoily, Abu Ashfaqur Sajib.

**Formal analysis:** Tamim Ahsan, Sabrina Samad Shoily.

**Methodology:** Tamim Ahsan.

**Supervision:** Abu Ashfaqur Sajib.

**Validation:** Tasnim Ahmed.

**Writing – original draft:** Sabrina Samad Shoily.

**Writing – review & editing:** Tasnim Ahmed, Abu Ashfaqur Sajib.

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
