## [Decision Letter · Decision Letter 0]

10 May 2023

PONE-D-23-03196Role of the redox state of the Pirin-bound cofactor on interaction with the master regulators of inflammation and other pathwaysPLOS ONE

Dear Dr. Sajib,

Thank you for submitting your manuscript to PLOS ONE. After careful consideration, we feel that it has merit but does not fully meet PLOS ONE’s publication criteria as it currently stands. Therefore, we invite you to submit a revised version of the manuscript that addresses the points raised during the review process.

We look forward to receiving your revised manuscript.

Kind regards,

Sheikh Arslan Sehgal, PhD

Academic Editor

PLOS ONE

Journal Requirements:

"This study was supported by a grant under the Special Allocation in Science and Technology from the Ministry of Science and Technology, Bangladesh to AAS. The authors are thankful for the support."

"This study was supported by a grant under the Special Allocation in Science and Technology from the Ministry of Science and Technology, Bangladesh"

3. Please ensure that you include a title page within your main document. We do appreciate that you have a title page document uploaded as a separate file, however, as per our author guidelines (http://journals.plos.org/plosone/s/submission-guidelines#loc-title-page) we do require this to be part of the manuscript file itself and not uploaded separately.

Could you therefore please include the title page into the beginning of your manuscript file itself, listing all authors and affiliations

"This study was supported by a grant under the Special Allocation in Science and Technology from the Ministry of Science and Technology, Bangladesh."          

Reviewers' comments:

Reviewer's Responses to Questions

**Comments to the Author**

1. Is the manuscript technically sound, and do the data support the conclusions?

Reviewer #1: Yes

2. Has the statistical analysis been performed appropriately and rigorously? 

Reviewer #1: Yes

3. Have the authors made all data underlying the findings in their manuscript fully available?

Reviewer #1: Yes

4. Is the manuscript presented in an intelligible fashion and written in standard English?

Reviewer #1: Yes

5. Review Comments to the Author

Reviewer #1: The information regarding different diseases and their possible cure mentioned in the article is well explained so I am going to recommend this article for publication. it's very well written. Only the future perspectives for further research in this specific aspect should also mentioned.

6. PLOS authors have the option to publish the peer review history of their article (what does this mean?). If published, this will include your full peer review and any attached files.

Reviewer #1: No

---

## [Author Response · Author response to Decision Letter 0]

5 Jun 2023

Responses to the academic editor

Response: We have updated the manuscript following PLOS ONE's style requirements.

"This study was supported by a grant under the Special Allocation in Science and Technology from the Ministry of Science and Technology, Bangladesh to AAS. The authors are thankful for the support."

"This study was supported by a grant under the Special Allocation in Science and Technology from the Ministry of Science and Technology, Bangladesh"

Response: We have removed the funding-related information from the Acknowledgements Section. The current Funding Statement is OK.

3. Please ensure that you include a title page within your main document. We do appreciate that you have a title page document uploaded as a separate file, however, as per our author guidelines (http://journals.plos.org/plosone/s/submission-guidelines#loc-title-page) we do require this to be part of the manuscript file itself and not uploaded separately.

Response: We have updated the title page according to the guidelines. It is now included at the beginning of the main manuscript. 

"This study was supported by a grant under the Special Allocation in Science and Technology from the Ministry of Science and Technology, Bangladesh." 

Response: We could not find the section for “Financial Disclosure” on the “Additional Information” page. We would like to mention the roles of the funders in the revised manuscript as "This study was supported by a grant under the Special Allocation in Science and Technology from the Ministry of Science and Technology, Bangladesh. The funders had no role in study design, data collection and analysis, decision to publish, or preparation of the manuscript". 

Response: We have reviewed the reference list following the suggestions. 

Responses to the reviewer 

1. The information regarding different diseases and their possible cure mentioned in the article is well explained so I am going to recommend this article for publication. it's very well written. Only the future perspectives for further research in this specific aspect should also mentioned.

Response: Thank you for your remarks on the manuscript. We have mentioned the future perspectives for future research in the revised manuscript (highlighted in yellow).

---

## [Editor Report · Decision Letter 1]

13 Jul 2023

Role of the redox state of the Pirin-bound cofactor on interaction with the master regulators of inflammation and other pathways

PONE-D-23-03196R1

Dear Dr. Sajib,

We’re pleased to inform you that your manuscript has been judged scientifically suitable for publication and will be formally accepted for publication once it meets all outstanding technical requirements.

Kind regards,

Sheikh Arslan Sehgal, PhD

Academic Editor

PLOS ONE
---

## [Editor Report · Acceptance letter]

17 Jul 2023

PONE-D-23-03196R1 

Role of the redox state of the Pirin-bound cofactor on interaction with the master regulators of inflammation and other pathways 

Dear Dr. Sajib:

I'm pleased to inform you that your manuscript has been deemed suitable for publication in PLOS ONE. Congratulations! Your manuscript is now with our production department. 

Kind regards, 

on behalf of

Dr Sheikh Arslan Sehgal 

Academic Editor

PLOS ONE